# Modulus and Strength of Concretes with Alternative Materials

**DOI:** 10.3390/ma13194378

**Published:** 2020-10-01

**Authors:** Ana Elisabete Paganelli Guimarães de Avila Jacintho, Ivanny Soares Gomes Cavaliere, Lia Lorena Pimentel, Nádia Cazarim Silva Forti

**Affiliations:** Exact Sciences, Environmental and Technologies Center, Pontifical Catholic University of Campinas (PUC-Campinas), Campinas 13086-099, Brazil; ivannycavaliere@gmail.com (I.S.G.C.); lialp@puc-campinas.edu.br (L.L.P.); nadia.cazarim@puc-campinas.edu.br (N.C.S.F.)

**Keywords:** concrete, natural aggregates, recycled concrete aggregates, waste porcelain aggregates, elasticity modulus

## Abstract

This paper presents a study with concretes produced with natural aggregates, recycled concrete aggregates (RCA) and waste porcelain aggregates (WPA). The study analyzed the influence of recycled aggregates in the mechanical properties of conventional concretes and evaluated the difference between measured and predicted values of elasticity modulus. The incorporation of WPA in concrete showed better mechanical results compared to the concretes produced with RCA. Measured elasticity moduli were lower than moduli predicted by NBR 6118:2014 and fib Model Code 2010, while measured results were greater than values predicted by Eurocode 2:2004 and ACI 318:2014, as expected, which indicated the safety of the latter two standards.

## 1. Introduction

Due to the exponential growth of the population, the construction industry has been highly demanded to attend society by building new constructions. Therefore, the construction industry is considered one of the main sectors with negative environmental impact, since it consumes huge amounts of non-renewable natural resources and generates an extensive volume of construction and demolition wastes [1,2,3].

In Brazil, it is estimated 45 million tons of construction and demolition waste (C&DW) are collected by the Brazilian municipalities annually, representing 58% of the total solid urban waste generated [4]. The incorrect disposal of these huge amounts of wastes has caused many environmental and social problems, as they compromise the urban landscape, contaminate the soil and contribute to clogging the urban drainage systems [2,5].

Instead of using natural aggregates in the production of new concrete, a viable alternative is using recycled aggregates obtained from construction and demolition waste [3,6]. Many studies recognize concretes composed of recycled aggregates as coarse and/or fine fractions, with similar mechanical properties to conventional concretes [7,8,9]. There was a study on an “ecological concrete” for structural use made by Brazilian and Italian researchers in 2004 [10]. The use of recycled aggregates in structural concretes can only be widely adopted by the construction industry if the concrete design parameters are well studied [11].

The elasticity modulus of concrete is a fundamental parameter in concrete structure design because it is possible to predict the structural behavior in service under load action and determine the deformations and displacements distribution [12,13].

To relate the experimental values with predicted values from standards, it is always necessary to validate the research and the standards indication, as the authors of [14] did with their columns of confined concrete. Several different prediction equations have been proposed to calculate concrete elasticity modulus from its compressive strength, but it is also possible to determine it by mechanical tests [11,13]. The elasticity modulus values obtained in experimental tests, as with other design parameters of concretes structures, differ from those predicted by empirical equations proposed by standards, even though the formulations consider graphs made up of empirical values [11,13,14,15,16,17,18].

These values cannot differ significantly, as a slight change in modulus translates to a slight difference in the deformation of a structure, and thus a lack of accuracy between experimental and predicted modulus can reflect in an under- or overdesign of concrete structures [13].

Accordingly, this study aimed to study the modulus of elasticity of concretes with compressive strength of 20 and 40 MPa with partial replacement rates (30% and 50%) of basaltic coarse aggregates by recycled concrete aggregates (RCA) and waste porcelain aggregates (WPA), as well as evaluate the difference between the values of measured and predicted modulus calculated by the Brazilian standard NBR 6118:2014 [19], fib Model Code 2010 [20], Eurocode 2:2004 [21] and ACI 318:2014 [22].

Thus, this study, in contrast to those mentioned above, showed the possible use of national and international standards for structural projects, with conventional strength concrete made with certain recycled aggregates.

## 2. Materials and Methods

The purpose of this research was to investigate the mechanical properties of concretes with moderate strength because moderate strength concretes are frequently used in reinforced concrete construction. The Brazilian standard NBR 8953:2015 [23] classifies these concretes in Group 1 with a characteristic compressive strength (f_ck_) ranging from 10 to 50 MPa.

Thus, two groups of concretes were produced: 20 MPa and 40 MPa concretes. For each group, different coarse aggregates were used with replacement rates of 30% and 50%: natural aggregates (NCA), that was basalt from Campinas, Brazil; recycled concrete aggregates (RCA), made by waste concrete from PUC Campinas laboratory; and waste porcelain aggregates (WPA), of a porcelain factory of electrical insulators in the city of Pedreira, Brazil.

The notation of all concretes was denominated as shown Figure 1. First, the compressive strength (20 or 40 MPa) is displayed, then the type of coarse aggregate incorporated (RCA or WPA) and finally the replacement rate (30% or 50%). Conventional concretes or reference concretes, with only natural aggregates, are named as C20 or C40 depending on the compressive strength group.

The present study followed the flowchart displayed in Figure 2. The waste materials were recycled, the employed materials were characterized, the concretes mix proportions were decided, the mixtures were cast in cylinder and cube specimens, the mechanical tests of specimens were carried out, the concretes’ mechanical properties were predicted from formulations proposed by standards, all the data were analyzed and the final considerations were concluded.

The specimen’s preparation and mechanical tests were executed at the Construction Materials and Structures Laboratory from Pontifical Catholic University of Campinas, Brazil.

### 2.1. Materials Selection and Characterization

For concretes, production used the following constituent materials: rapid hardening cement (Brazilian type cement CP V ARI, LafargeHolcim, Santo André, Brazil) with a density of 3.13 g/cm^3^; fine natural aggregate (FNA) that was quartzite sand from Campinas, Brasil; coarse natural aggregate (CNA) that was basalt from Campinas, Brazil; recycled concrete aggregates (RCA), made by waste concrete from PUC Campinas laboratory; waste porcelain aggregates (WPA), of a porcelain factory of electrical insulators in the city of Pedreira, Brazil; plasticizer, from GCP Applied Technology, Sorocaba, Brazil and water from Campinas city. The CNA was basaltic gravel, whereas the FNA was quartzite sand, both of which are the most widely-used aggregates in the state of São Paulo, Brazil.

Two plasticizers were employed in the concretes: a water reducer from GCP Applied Technology, Sorocaba, Brazil and water from Campinas city for the 20 MPa concretes and a polycarboxylic ether-based superplasticizer, from GCP Applied Technology, Sorocaba, Brazil and water from Campinas city for the 40 MPa concretes. The plasticizer’s producers suggest a dosage between 0.6% and 1.0% upon cement consumption when using the water reducer plasticizer and a dosage between 0.15% and 0.80% when using the superplasticizer.

The recycling process of concrete and porcelain waste consisted of crushing the porcelain into small pieces using a hammer and then crushing the small pieces in a jaw crusher (Contenco Industry, São José da Lapa, Minas Gerais, Brazil). Four samples of 5 kg of each waste were crushed. The samples were bent for the particle size analysis to determine a grading curve for the recycled coarse aggregates as similar as possible to the natural coarse aggregates. Figure 3 exhibits the whole process of recycling.

During the crushing process of concrete waste, it was seen that some gravels were detached from the mortar of the concrete waste; thus, some gravels returned to their original natural aggregate, but still surrounded by a fine old cement paste. In the porcelain waste crushing, great care was required during handling due to the splinters caused by breaking it.

Figure 4 illustrates a visual difference between natural and recycled aggregates. The physical properties of natural and recycled aggregates are assembled in Table 1, including the standards consulted to realize the experiments. Figure 5 shows the fine aggregates distribution curves and Figure 6 shows the coarse aggregates distribution curves with their respective limits implied by NBR NM 248:2003 [24].

### 2.2. Mix Design Proportions and Specimen Preparation

Table 2 lists the concrete proportions. The mix design proportions were based on the modified IPT method [25].

In particular, the natural coarse aggregates (NCA) were volumetrically replaced with recycled coarse aggregates (RCA and WPA) with replacements rates of 0%, 30% and 50%. The quantity of both recycled coarse aggregates was adjusted according to the relation of recycled aggregates density to natural aggregates density.

Furthermore, 0.60 water/cement ratio was employed for 20 MPa reference concretes and 0.40 for 40 MPa reference concretes. For concretes containing recycled aggregates, the water/cement ratios were reduced, expecting a possible reduction of their mechanical properties. In the case of 40 MPa concretes, the water/cement ratio remained the same because of the different plasticizer used: a water reducer plasticizer was used for 20 MPa concretes, while 40 MPa concretes used a superplasticizer. At the moment of molding the specimens with 40 MPa concrete, there was not enough plasticity to mold the specimens adequately and the prior plasticizer adopted (water reducer plasticizer) had to be changed to a different one (superplasticizer).

The Brazilian standard NBR 15116:2004 [26] prescribes the inclusion of pre-soaked water in concretes with RCA as RCA diminishes water absorption and slightly reduces their mechanical properties. Therefore, for RCA concretes, 60% pre-soaked water over RCA water absorption capacity was inserted. For WPA concretes, it was decided not to adopt pre-soaked water as porcelain water absorption capacity is too low.

The concretes mix procedure pursued a sequence of events. First, the concrete mixer was moistened. The kneading water was separated into two portions: in the first portion, two parts of plasticizer was incorporated, and, in the second portion, only one part was incorporated. As plasticizer total amount was divided into four parts, the last part was saved in case it would not be possible to mold the specimens. Subsequently, all coarse aggregates and the first portion of kneading water was added, the cement was also included and those present materials were mixed. The fine natural aggregates and the second portion of kneading water were added, and, finally, the whole mixture was mixed for 5 min. At last, a slump test was performed only to investigate if mortar content was adequate, and, in the case it was not, new parts of plasticizer were carefully added considering the producer specified dosage. Figure 7 illustrates how materials were separated; the two blue buckets represent the kneading water and the red bucket represents the pre-soaked water.

### 2.3. Mechanical Testing

Concrete specimens were cast, while a vibrating table was used to compact the concrete. For each concrete mixture, thirty 100 mm × 200 mm cylindrical specimens and six 100 mm cubic specimens were prepared. In particular, cylindrical specimens were used to measure the compressive strength (f_cm,cyl_) (six specimens for 7 days old and six specimens for 28 days old), the elasticity modulus (E_cm_) (six specimens for 7 days old and six specimens for 28 days old) and the splitting tensile strength (f_ctm,sp_) (six specimens for 28 days old). Cubic specimens were used only to test compressive strength at 28 days old and compare with values of compressive strength tested with cylindrical specimens.

The values of measured elasticity modulus were associated with initial tangent modulus; hence, the measured initial tangent modulus must be compared to the predicted tangent modulus.

### 2.4. Analysis of Variance (ANOVA)

Single-factor ANOVA was developed by Fisher (1890–1962) and consists of observing possible differences between two or more samples averages at a 5% level of significance [27]. ANOVA’s response is obtained from hypothesis testing that can be performed by *p*-value. *P*-value is designated as the probability of any sample average being more distant than the other samples’ average. *P*-value evaluation is given as:When the *p*-value is greater than or equal to the level of significance, there is no significant difference.When the *p*-value is lower than the level of significance, there is indeed a significant difference between the samples.

ANOVA was used to evaluate the influence of three levels of replacement rates of recycled aggregates in compressive strength and elasticity modulus of all concretes. It was also applied to the investigation of geometry changing influence in compressive strength, by using cubic and cylindrical specimens.

### 2.5. Prediction Formulations

The parameters elasticity modulus and tensile strength were predicted using equations proposed by consulted standards NBR 6118:2014 [19], fib Model Code 2010 [20], Eurocode 2:2004 [21] and ACI 318:2014 [22]. Both parameters are predicted from concrete compressive strength. NBR 6118:2014 [19], fib Model Code [20] and ACI 318:2014 [22] use characteristic compressive strength (f_ck_) in the calculation, while Eurocode 2:2004 [21] uses measured compressive strength (f_cm_).

Deviation values prescribed by the standards were subtracted from f_cm_ to determine f_ck_. As each standard assumes a different deviation value, different values of f_ck_ were determined. Standard deviation refers to concrete preparation conditions that assume laboratory conditions at the time of molding as well as how materials were separated. Table 3 lists the f_ck_ calculated for all concretes considering each standard deviation.

Brazilian standard NBR 6118:2014 [19] considers a deviation of 6.60 MPa for concretes with strength ranging from 20 MPa to 90 MPa while Eurocode 2:2004 [21] considers a bigger deviation of 8.0 MPa for concretes with a range of strength between 12 MPa and 90 MPa. American standard ACI 301:2010 [28] considers deviation of 8.274 MPa for concretes with strength ranging from 20 MPa to 35 MPa, while, for concretes with strength above 35 MPa, Equation (1) is recommended. Fib Model Code 2010 [20] suggests an 8 MPa deviation for concretes with compressive strength ranging from 12 MPa to 120 MPa.
(1)$$f_{c}^{\prime }=\frac{\left(f_{cm}-4.826\right)}{1.10}$$

#### 2.5.1. Elasticity Modulus Prediction

Table 4 presents the prediction formulations to predict the elasticity modulus of concretes proposed by four different concrete designing standards [19,20,21,22]. In general, the elasticity modulus is obtained from concrete compressive strength, coarse aggregate nature and concrete density. The equations proposed by NBR 6118:2014 [19] and fib Model Code 2010 [20] are similar to each other, as both consider the parameter α_E_, which depends on the nature of coarse aggregate.

The parameter related to coarse aggregate nature decreases or increases the predicted value of elasticity modulus: if the coarse aggregate is basalt or diabase then the value of elasticity modulus increases 20%; if it is granite, gneiss or quartzite the value remains constant; if the coarse aggregate used is limestone, the elasticity modulus value reduces 10%; and if it is sandstone coarse aggregate, it reduces 30%. In this research, the natural coarse aggregates used to produce the concretes were basaltic, thus the value of α_E_ is equal to 1.20.

According to Brazilian standard NBR 8522:2017 [29], which determines the measured elasticity modulus, tangent modulus (or elasticity modulus) can be defined as the slope of the line tangent to the stress–strain curve between σ_a_ and 30% of f_cm_, while secant modulus (or deformation modulus) is the slope of the stress–strain curve between σa and a stress within plastic strain zone (above 30% of f_cm_).

Differently, fib Model Code 2010 [19] and Eurocode 2:2004 [21] set for the elasticity modulus a 40% limit of f_cm_, and, over this limit, a reduced modulus can be obtained (secant modulus). ACI 318:2014 [22] defines elasticity modulus as the slope of the line drawn from a stress of zero to compressive stress of 45% of f_cm_, and it does not propose an equation to predict secant modulus.

#### 2.5.2. Tensile Strength Prediction

Table 5 lists the prediction equations to predict the direct tensile strength (f_ct_) and splitting tensile strength (f_ct,sp_). NBR 6118:2014 [19], fib Model Code 2010 [20] and Eurocode 2:2004 [21] propose the same equation to calculate f_ct_ while ACI 318:2014 [22] only presents an equation to predict f_ct,sp_.

The f_ct,sp_ is predicted in the same manner by Eurocode 2:2004 [21] and NBR 6118:2014 [19]; both standards consider a conversion factor of 0.90 to predict the splitting tensile strength out of direct tensile strength. Fib Model Code 2010 [20], on the other hand, considers a conversion factor of 1.00.

## 3. Results

The measured parameters were obtained by mechanical testing (i.e., compressive strength, tensile strength and elasticity modulus). Compressive strength was measured to predict the elasticity modulus and tensile strength values and to compare compressive strength using cylindrical and cubic specimens. Elasticity modulus and splitting tensile strength were measured to be compared with their predicted values.

### 3.1. Measured Mechanical Results

#### 3.1.1. Compressive Strength, f_cm_

Table 6 presents a tested 28-day average compressive strength (f_cm,cyl_) molded in six cylindrical specimens (100 mm × 200 mm). Individual values were discarded in respect of 6% COV (coefficient of variation).

The individual specimen results of C40-RCA50 and C40-WPA30 were very dispersive, only three results of six specimens were considered for the average and the COV calculation.

The influence of recycled aggregates on concrete compressive strength was analyzed by using ANOVA single factor. ANOVA indicated which recycled aggregate concrete was significantly different from reference concrete (conventional concrete). Table 7 presents a one-way ANOVA analysis of the average compressive strength of 20 MPa and 40 MPa concretes. To determine if the compressive strength of a concrete with recycled aggregates differed from its reference concrete, ANOVA analysis was applied. Each row of Table 7 compares a pair of data. In these analysis results, *p*-value < 0.05 indicates the compressive strengths have a significant difference. When *p*-value ≤ 0.05, the test indicates the values are similar. Columns F and f are ANOVA auxiliary variables [30].

The test of C20 and C20-RCA30 indicated that their compressive strengths are statistically different (*p*-value = 0.0200 < 0.05) the average compressive strength reduced 4.06 MPa due to the addition of recycled aggregates. On the other hand, the test of C40 and C40-WPA50 indicated the compressive strength are not significantly different (*p*-value = 0.17). Even though the average difference is 6.54 MPa, the distribution of data is such that is t is not possible to determine any influence from the addition of recycled aggregates.

Figure 8 illustrates the influence of RCA and WPA incorporation with replacement rates of 30% and 50%. The 20 MPa concretes are in dark gray columns while the 40 MPa concretes are in light gray columns. A red line was drawn from the f_cm,cyl_ of reference concretes (C20 and C40) to compare with f_cm,cyl_ of recycled aggregate concretes (with RCA and WPA).

RCA concrete compressive strength was indicated by ANOVA analyses to differ significantly from reference compressive strength, while WPA concrete compressive strength was similar to reference compressive strength. Therefore, it was indicated that RCA incorporation negatively affects the strength of 20 MPa concretes, while WPA is feasible to be used as aggregates.

Compressive strength decrease escalated when RCA replacement rate was increased because, by increasing the RCA replacement rate, compressive strengths of concretes were reduced due to RCA characteristics such as heterogeneity, porosity, high absorption and low strength [31]. At the age of 28 days, the compressive strength of C20-RCA30 decreased 11.25% while that of C20-RCA50 decreased 36.17%.

The compressive strength of WPA concretes was equal to the conventional concrete compressive strength. It was assumed by Campos and Paulon [32] that this is due to the chemical and physical similarities between waste porcelain aggregates and natural aggregates. It was also state by Ferreira, et al. [33] that porcelains have high mechanical strength as a result of the presence of clay, feldspar and quartz, and therefore the mechanical results of WPA concretes are very satisfactory. For 40 MPa concretes, neither of the recycled aggregates used in this research (RCA and WPA) affected substantially the compressive strength, as indicated by ANOVA analysis.

#### 3.1.2. Compressive Cubic Strength, f_cm,cubic_

The average compressive cubic strength (f_cm,cube_) was measured from six cubic specimens of 100 mm. Table 6 presents the average compressive cubic strength in comparison to average compressive cylindrical strength. Correspondingly, individual values were discarded and 6% of COV (coefficient of variation) was respected.

It is understood that the size and shape of specimens vary according to which standard is adopted and which types of materials are used at the time of testing. Commonly, European countries use cubic specimens to determine compressive strength, while in Brazil, USA and other countries cylindrical specimens with 2.00 height/diameter ratio are often used. Some countries use both cubic and cylindrical specimens [34,35].

According to Watanabe, et al. [36], the stress–strain curves of concrete strongly depend on the aspect ratio of the concrete specimen. Specimen parameters such as size, geometry and humidity conditions generally affect the results obtained in mechanical tests [37]. It was appointed by Kaish et al. [38] that these effects are deeply known in tensile strength results while in compressive strength the effects of size and shape of concrete specimens are not widely investigated.

Therefore, this study aimed to evaluate the compressive strength behavior of concretes cast in cylindrical and cubic specimens. An ANOVA analysis was performed to validate the difference between these two strengths. Table 8 presents an ANOVA analysis comparing compressive strength cast in cylindrical specimens (f_cm,cyl_) and cast in cubic specimens (f_cm,cube_) of all concretes.

Figure 9 illustrates the difference between f_cm_ and f_cm,cube_. Mucciacia et al. [37] explained that the height/diameter ratio is responsible for this difference: the higher is the height/diameter ratio, the lower is the strength. The height/diameter ratio used in this research was 1.00 for cubic specimens and 2.00 for cylindrical specimens.

For 20 MPa concretes, cubic strengths were higher than cylindrical strengths (with an increase ranging from 11.61% to 53.02%). All results were validated by an ANOVA analysis, which indicated that the type of specimen significantly influenced the strength result.

It was said by Sinaie, et al. [39] that specimens with a lower height/diameter ratio normally lead to higher strengths, and Neville [40] indicated that, when the height/diameter ratio decreases, stresses and peak deformations usually also reduce, leading to greater strengths. It was explained by Gyurkó and Nemes [41] that a possible reason would be the effect of enclosing caused by the press plates all over the cube’s height, but, in cylinders, it does not reach some part of the height.

Unlike 20 MPa concretes, 40 MPa concretes did not have a significant difference between cylindrical and cubic strengths. The type of geometry did not substantially affect the 40 MPa concrete strength results, except for C40-WPA30, which indicated a significant difference. According to Gyurkó and Nemes [41], the relationship between the cylinder and cube strengths increases greatly with strength classification increasing. The Soares at al. [42] also found in their investigation that the effect of specimen geometry is more significant in low strength class concrete (C20/25).

Table 9 shows the relationship between the cylindrical and cubic strength of concretes, and it was observed that the cylindrical/cubic strength ratio of 40 MPa concretes is greater than that of 20 MPa concretes.

#### 3.1.3. Splitting Tensile Strength, f_ctm,sp_

Average splitting tensile strength was determined by measuring six cylindrical specimens (200 mm × 100 mm) at the age of 28 days. Table 6 presents all concretes values of splitting tensile strength. Individual values were discarded if they favored the increase of COV (coefficient of variation) above 6%.

Compressive strength is the main mechanical property of concrete used by many engineers for designing and inspecting concrete structures. Splitting tensile strength and flexural tensile strength, on the other hand, are not so commonly used; normally they are 10% and 15% of compressive strength, respectively [35]. As shown in Table 10, the splitting tensile strength of the concretes is from 8% to 12% of the compressive strength.

It was also state by Mehta and Monteiro [35] that compressive and tensile strength are closely related: as compressive strength increases, tensile strength also increases, but at a decreasing rate. Thus, the higher is the compressive strength, the lower is the tensile/compressive strength ratio.

This statement is proven by analyzing the results in Table 10, which shows that the concrete with the highest strength (C40-WPA50) is the one with the lowest tensile/compressive strength ratio, while the concrete with the lowest strength (C20-RCA50) has the lowest tensile/compressive strength ratio.

#### 3.1.4. Elasticity Modulus E_cm_

Six cylindrical specimens (100 mm × 200 mm) were tested at 28 days to obtain an average elasticity modulus. The test followed NBR 8522:2017 [29], thus it was necessary to discard some individual values when effective strength (f_c,ef_) was different from concrete compressive strength by ±20%. COV (coefficient of variation) of 6% was considered. Table 6 displays the results of the average elasticity modulus.

ANOVA single factor was used to investigate which recycled aggregate (RCA or WPA) and in what proportions significantly affected the elasticity modulus. Reference concrete elasticity moduli were individually compared to those of recycled aggregate concretes. Table 11 exhibits the ANOVA analysis of concretes elasticity modulus.

Figure 10 illustrates a comparison between elasticity modulus of conventional concretes and elasticity modulus of recycled aggregate concretes. The 20 MPa concretes are in dark gray columns, while the 40 MPa concretes are in light gray columns. A red line was drawn from the elasticity modulus of conventional concretes (C20 and C40) to compare with the elasticity modulus of RCA and WPA concretes.

Through ANOVA analysis, it was possible to establish that both recycled aggregates (RCA and WPA) affect the elasticity modulus because concretes with recycled aggregates indicated significant difference when compared to reference concretes, except for C40-RCA50, for which, according to ANOVA analysis, the average elasticity modulus is similar to the average elasticity modulus of conventional concretes.

Recycled aggregates typically reduce the value of concretes elasticity modulus. It was shown by Manzi, et al. [43] that, as RCA replacement rate increases, the modulus of elasticity decreases, because concrete elasticity modulus is influenced by porosity presented in each of the concretes parts.

However, in this study, the concretes with 30% of RCA (C20-RCA30 and C40-RCA30) had a respective increase of 5.84% and 4.87% in the elasticity modulus, while the concrete with 50% of RCA had a reduction. The Milhomen et al. [44] also observed an increase in the modulus of elasticity of self-consolidating concrete produced with both recycled fine and coarse aggregates. The authors believed that the improvement was due to better adhesion of cement paste and mortar to the particles of recycled aggregates.

Regarding the use of waste porcelain aggregates (WPA), some authors verified that the characteristics of high rigidity and high mechanical strength of porcelain benefit concrete mechanical properties [32,45]. WPA concrete tended to increase the elasticity modulus of 20 MPa and 40 MPa concretes, with both replacement rates (30% and 50%), in the range of 15.02–22.93%.

It is believed that the cause of this increase is due to the rough surface of the porcelain aggregates (non-polished surface). When De Argollo Ferrão, et al. [45] analyzed the aggregate–paste transition zone with Scanning Electron Microscopy (SEM), they observed that the present roughness in porcelain without polishing allows the paste to adhere easily, thus benefiting the modulus of elasticity.

### 3.2. Predicted Values

#### 3.2.1. Predicted Tensile Strength

Table 12 lists the values of direct tensile strength (f_ct_) and splitting tensile strength values (f_ct,sp_) calculated by standards NBR6118:2014 [19], fib Model Code 2010 [20], Eurocode 2:2004 [21] and ACI318:2014 [22] in comparison to measured results of splitting tensile strength (f_ctm,sp_).

Different values of tensile strength were obtained through different standards calculations because they adopt different coefficients and deviation values. In general, the predicted values of splitting tensile strength of all concretes calculated by Eurocode 2:2004 [21] were lower than the rest of the values predicted by other standards.

For all concretes, values of splitting tensile strength predicted by ACI 318:2014 [22] were superior to values predicted by other standards, followed by values calculated by NBR 6118:2014 [19] and fib Model Code 2010 [20].

The 20 MPa concretes resulted in lower predicted splitting tensile strength when compared to the 40 MPa concretes, proving that the value of splitting tensile strength depends on the variation of concrete compressive strength.

As Brazilian standard NBR 6118:2014 [19] and fib Model Code 2010 [20] consider a correction factor of 0.90, values of predicted direct tensile strength (f_ct_) were lower than predicted splitting tensile strength (f_ct,sp_). Eurocode 2:2004 [21], on the other hand, considers a correction factor of 1.00, hence f_ct_ values were the same as f_ct,sp_ values.

The ratio between measured splitting tensile strength and measured compressive strength (f_ctm,sp_/f_cm,cyl_) was calculated. The ratio between predicted splitting tensile strength and characteristic compressive strength (f_ct,sp_/f_ck_) was also determined. Table 13 lists the measured and predicted ratios of these two parameters for all concretes.

#### 3.2.2. Predicted Elasticity Modulus

Table 14 brings together the predicted values of elasticity modulus for 20 MPa and 40 MPa concretes with equations proposed by NBR 6118:2014 [19], fib Model Code 2010 [20], Eurocode 2:2004 [21] and ACI318:2014 [22].

The predicted values of elasticity modulus resulted in different values when calculated by the different equations proposed by each standard because they consider different coefficients and deviation values. In general, the predicted elasticity modulus of all concrete (20 MPa and 40 MPa concretes) calculated by ACI 318:2014 [22] were lower than those predicted by other standards.

For 20 MPa concretes, the elasticity moduli predicted by fib Model Code 2010 [20] were superior to those predicted by other standards and were followed by the moduli calculated by NBR 6118:2014 [19] and Eurocode 2:2004 [21]. Except for C20-RCA50, the descending order of predicted moduli were calculated by: fib Model Code 2010 [20], Eurocode 2:2004 [21], NBR 6118:2014 [19] and ACI 318:2014 [22].

For 40 MPa concretes, the predicted elasticity moduli calculated by NBR 6118:2014 [19] were similar to the modulus predicted by fib Model Code 2010 [20], although the moduli with the highest occurrence of great values were those calculated by fib Model Code 2010 [20].

The 20 MPa concretes resulted in lower predicted elasticity modulus when compared to the 40 MPa concretes, proving that the value of elasticity modulus depends on the variation of concrete compressive strength.

The consulted standards do not indicate a value which represents the nature of recycled aggregate of concrete (RCA) and porcelain (WPA); therefore, the parameter depending on the nature of coarse aggregate (α_E_ = 1.20) used was the same for all calculations, even though the predicted modulus of elasticity was similar to measured compressive strength behavior.

### 3.3. Comparison between Predicted Values and Measured Mechanical Results

#### 3.3.1. Predicted vs. Measured tensile strength

Figure 11 illustrates the difference between measured and predicted splitting tensile strength of all concretes. Measured splitting tensile strength (f_ctm,sp_) are indicated in green columns while predicted splitting tensile strength (f_ct,sp_) calculated by NBR6118:2014 (gray) [19], fib Model Code 2010 (blue) [20], Eurocode 2:2004 (yellow) [21] and ACI 318:2014 (orange) [22] are shown in colored columns.

For 20 MPa concretes, most measured splitting tensile strength were lower than predicted values calculated by ACI 318:2014 [22], indicating a lack of safety by the American standard. On the contrary, for 40 MPa concretes, all measured splitting tensile strengths were greater than splitting tensile strengths predicted by all the standards.

#### 3.3.2. Predicted vs. Measured Elasticity Modulus

The comparative analysis of measured elasticity modulus and predicted elasticity modulus calculated by standards is illustrated in Figure 12. Measured elasticity modulus is indicated in black columns while predicted elasticity modulus are indicated in different colored columns in the figure.

The difference between measured and predicted elasticity modulus is more alarming when elasticity modulus was calculated by fib Model Code 2010 [20] and NBR 6118:2014 [19]. Both standards did not indicate the safety of structure serviceability. Safety is obtained when the measured modulus is equal to or greater than the predicted modulus. For conventional concretes (C20 and C40), the measured elasticity modulus was lower than modulus predicted by fib Model Code 2010 [20] (in 8.59% and 13.45%, respectively) and by NBR 6118:2014 [19] (in 0.82% and 13.62%, respectively).

ACI 318:2014 [22] and Eurocode 2:2004 [21], on the other hand, indicate safety for conventional concretes because measured elasticity modulus was superior to the predicted modulus. The measured elasticity moduli of C20 and C40 were 6.54% and 1.98% higher than the moduli predicted by Eurocode 2:2004 [21] and 45.98% and 26.95% higher than the moduli predicted by ACI 318:2014 [22].

For concretes with 50% of RCA, the measured elasticity modulus was inferior to the modulus predicted by fib Model Code 2010 [20]. The elasticity modulus of C20-RCA50 was 6.02% lower than the modulus predicted by fib Model Code 2010 [20], while that of C40-RCA50 was 11.01% lower than the modulus predicted by fib Model Code 2010 [20]. For concretes with 30% ARC, the measured modulus was superior to the predicted modulus. The measured elasticity modulus of C20-RCA30 was 7.36% larger than the fib Model Code 2010 [20] modulus, 20.55% larger than the NBR 6118:2014 [19] modulus, 24.68% larger than the Eurocode 2:2004 [21] modulus and 78.33% larger than the ACI 318:2014 [22] modulus. The measured elasticity modulus of C40-RCA30 is larger than modulus predicted by fib Model Code 2010 [20], NBR 6118:2014 [19], Eurocode 2:2004 [21] and ACI 318:2014 [22] by 1.22%, 2.14%, 19.07% and 49.95%, respectively.

The measured elasticity modulus of concretes with 30% and 50% of WPA were superior to the predicted modulus, except for C40-WPA50 with lower measured modulus than modulus predicted by fib Model Code 2010 [20] (4.03%) and by NBR 6118:2014 [19] (at 7.01%).

Another analysis was performed to calculate the modulus of elasticity considering different parameters related to aggregates nature (α_E_), prescribed by Brazilian standard NBR 6118:2014 [19]. The parameter α_E_ = 1.20 represents aggregates of basalt or diabasium, α_E_ = 1.00 represents granite aggregate or gneiss, α_E_ = 0.90 considers limestone aggregate and α_E_ = 0.70 indicates sandstone aggregate. For this study, α_E_ = 1.20 was initially considered in the calculations due to the use of basaltic coarse aggregates.

Considering α_E_ = 1.20, the concretes which resulted in a measured elasticity modulus lower than the predicted modulus were C20, C40, C40-RCA50 and C40-WPA50. Therefore, the elasticity modulus was calculated considering α_E_ = 1.00 (granite or gneiss) to investigate whether the measured elasticity modulus of these concretes would be above or below the values of modulus considering α_E_ = 1.00. Figure 13 shows the measured elasticity modulus in comparison to predicted modulus considering different parameters (α_E_).

Measured elasticity modulus (green columns) of C20, C40, C40-RCA50 and C40-WPA50 when predicted considering α_E_ = 1.00 (granite) were superior to the predicted modulus.

## 4. Conclusions

In the literature, it is acknowledged that conventional concrete has a complex structure. Knowing all the influence factors which affect concrete mechanical properties is recognized to be an arduous task. The insertion of recycled aggregates in concrete structure makes the task even more complex because of the great variability these aggregates causes in concrete structure behavior.

Most research on the usage of recycled aggregates is concerned with the influence of these aggregates on compressive strength. However, to ensure good quality structural concrete, it is important to investigate other parameters such as the elasticity modulus and tensile strength.

The elasticity modulus, associated with the control of structural deformations, can be measured by mechanical tests or predicted by equations proposed by standards. Hence, this research consisted of investigating the influence of recycled concrete aggregates (RCA) and waste porcelain aggregates (WPA) in mechanical properties of 20 MPa and 40 MPa strength class concretes, verifying if there are any similarities between the predicted values calculated by different standards, comparing the predicted values with measured results and evaluating the compressive strength of concretes cast in cylindrical and cubic specimens.

To validate the results, an ANOVA analysis was performed to understand the significant differences between the averages of measured results. In the compressive strength test, it was found that recycled concrete aggregates (RCA) affected the compressive strength of 20 MPa concretes but did not affect the compressive strength of 40 MPa concretes. Waste porcelain aggregates (WPA), on the other hand, did not affect the compressive strength of 20 MPa or 40 MPa concretes. It is believed that this is due to the physical and chemical similarities porcelain waste has with natural aggregates [32].

In the measured elasticity modulus investigation, it was found that RCA and WPA significantly affected the elasticity moduli of both 20 MPa and 40 MPa concretes. In this case, the concrete elasticity modulus with 30% of RCA was superior to the reference concrete modulus, which can be justified by the better adhesion between the cement paste and the mortar adhered to the particles of the recycled aggregates.

The elasticity modulus of concretes with WPA was higher than the reference modulus of elasticity. It is believed that the cause of this increase is due to the rough surface of non-polished porcelain that allows the paste to adhere easily and thus benefit the modulus of elasticity.

The elasticity modulus and splitting tensile strength calculated by the consulting standards [19,20,21,22] resulted in different values because each standard recommends different values of coefficients and deviation. In general, the decreasing order of predicted elasticity modulus is those calculated by: fib Model Code 2010 [20], NBR 6118:2014 [19], Eurocode 2:2004 [21] and ACI 318:2014 [22]. The decreasing order of predicted splitting tensile strength is calculated by: NBR 6118:2014 [19], fib Model Code 2010 [20], ACI 318:2014 [22] and Eurocode 2:2004 [21].

The measured elasticity moduli of conventional concretes (C20 and C40) were lower than the predicted moduli calculated by fib Model Code 2010 [20] and NBR 6118:2014 [19], indicating a lack of safety of serviceability from these standards. On the contrary, the measured elasticity modulus from conventional concretes was superior to those predicted by ACI 318:2014 [22] and Eurocode 2:2004 [21].

Measured splitting tensile strength results of 40 MPa concretes were superior to all predicted values, indicating safety from all the consulting standards. On the contrary, for most 20 MPa concretes, measured splitting tensile strength was lower than the values predicted by ACI 318:2014 [22], indicating a lack of safety by the American standard.

Different values of α_E_ were considered in modulus calculations, and it was found that, for concretes with measured elasticity modulus lower than the modulus predicted with α_E_ = 1.20 (basalt), when it was predicted with α_E_ = 1.00 (granite), the modulus was above them as expected.

An ANOVA analysis was also performed to verify whether the change in geometry of the specimens affects the compressive strength of the studied concretes. The geometry changes substantially affected the compressive strength of 20 MPa concrete but did not affect the strength of 40 MPa concrete. The results found in this research indicate that the effect of specimen geometry is more significant in low-strength concretes.

In 20 MPa concretes, the compressive strength of concrete cast in cubic specimens was superior to the compressive strength of concretes cast in cylindrical specimens. A possible cause would be the effect of containing the press plates that extend throughout the height of the cubes, but in the cylinders ends up not reaching part of the height.

## Figures and Tables

**Figure 1 materials-13-04378-f001:**
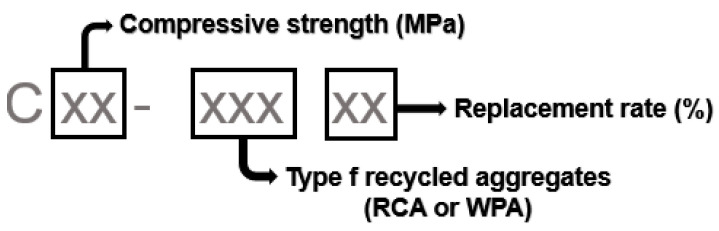
Concrete compositions classification.

**Figure 2 materials-13-04378-f002:**
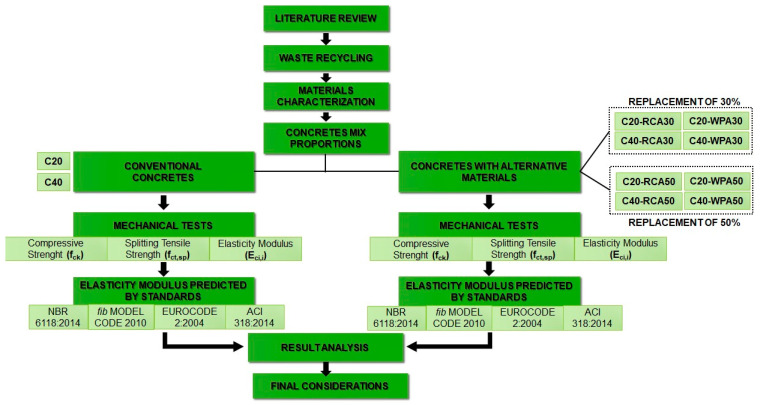
Research activities flowchart.

**Figure 3 materials-13-04378-f003:**
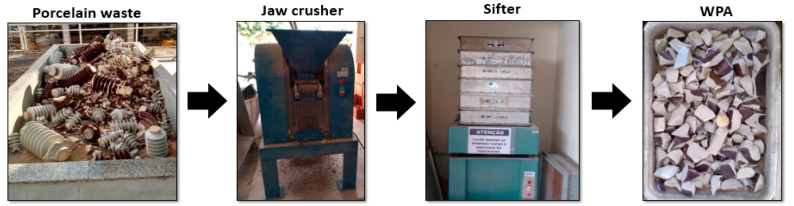
Recycling process of electrical porcelain insulators.

**Figure 4 materials-13-04378-f004:**
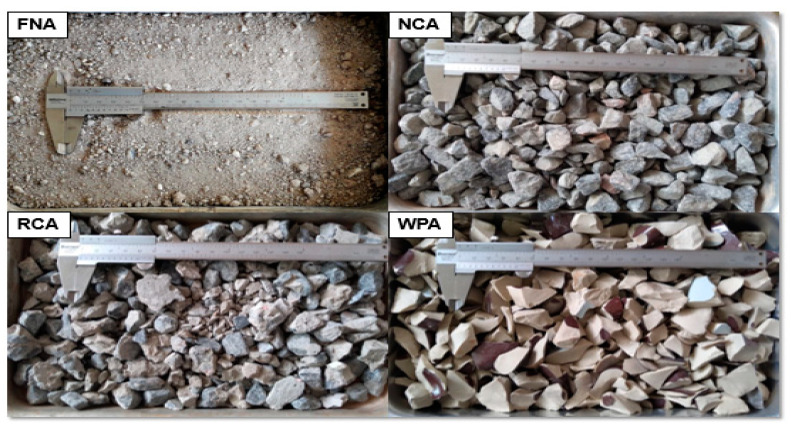
Comparison between natural and recycled aggregates.

**Figure 5 materials-13-04378-f005:**
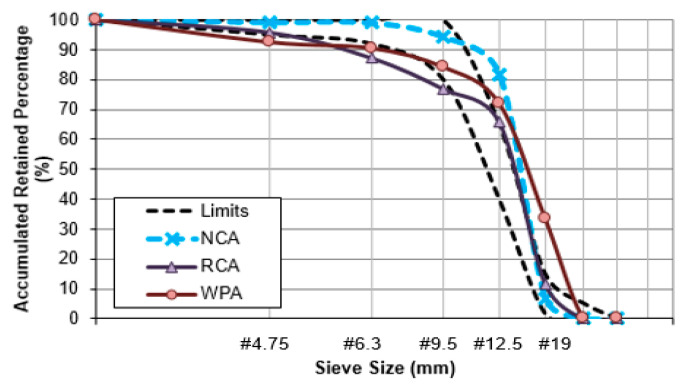
Gradation curves of recycled concrete aggregates (RCA) and waste porcelain aggregates (WPA) in comparison to natural coarse aggregate (NCA).

**Figure 6 materials-13-04378-f006:**
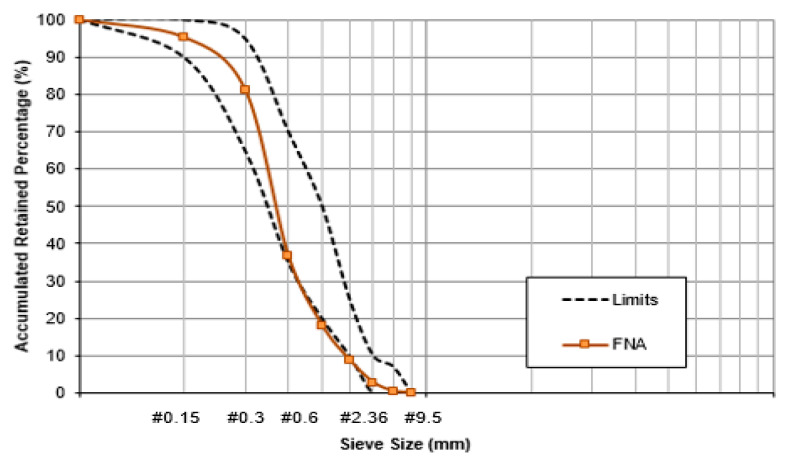
Gradation curves of fine natural aggregates (FNA).

**Figure 7 materials-13-04378-f007:**
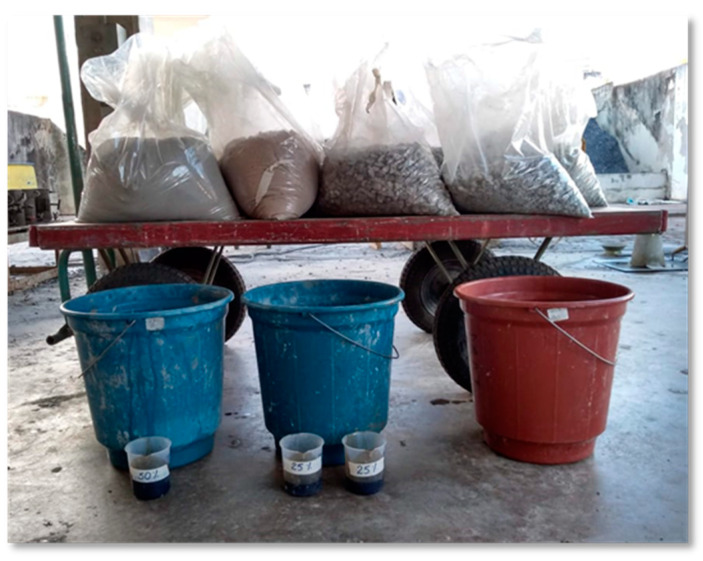
Materials separation for concrete production.

**Figure 8 materials-13-04378-f008:**
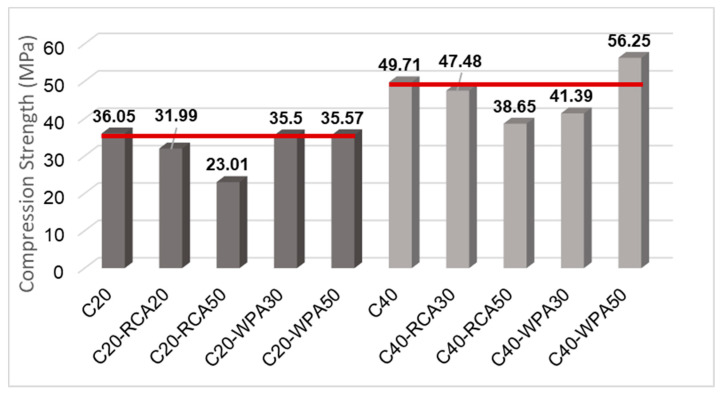
Compressive strength analysis of all concretes.

**Figure 9 materials-13-04378-f009:**
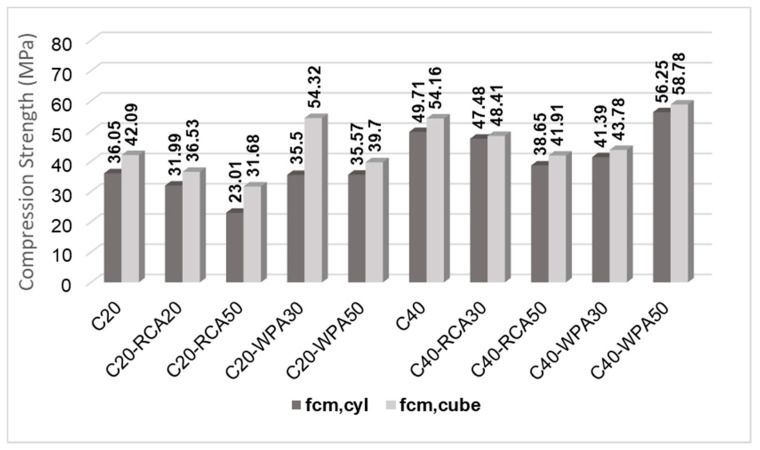
Cubic and cylindrical compressive strength comparative analysis of all concretes.

**Figure 10 materials-13-04378-f010:**
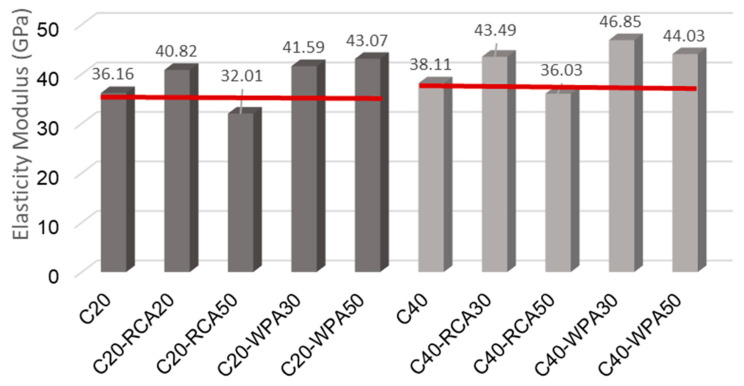
Elasticity modulus analysis of all concretes.

**Figure 11 materials-13-04378-f011:**
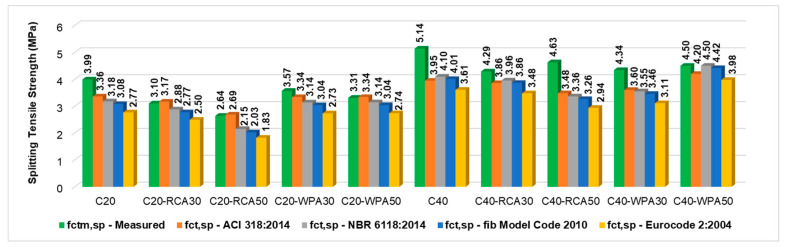
Comparison of measured and predicted splitting tensile strength considering standards.

**Figure 12 materials-13-04378-f012:**
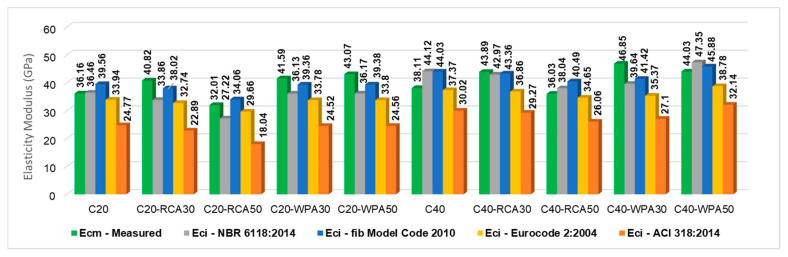
Comparison of measured and predicted elasticity modulus of concretes considering standards.

**Figure 13 materials-13-04378-f013:**
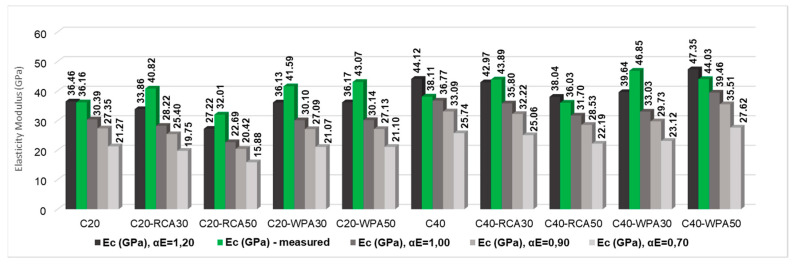
Comparison of measured elasticity modulus and elasticity modulus predicted with different values of α_E_.

**Table 1 materials-13-04378-t001:** Properties of natural and recycled aggregates.

Properties	Natural Aggregates		Recycled Aggregates	
	NFA	NCA	RCA	WPA
Particle size (mm)	4.75	25.00	25.00	25.00
Fineness modulus	2.43	6.97	6.63	1.87
Density (g/cm^3^)	2.53	2.74	2.39	2.34
Loose bulk density (g/cm^3^)	1.50	1.46	1.22	1.21
Compacted bulk density (g/cm^3^)	1.64	1.50	1.30	1.29
Void index rate (%)	ND	99.95	99.95	99.95
Water absorption rate (%)	ND	0.68	6.81	0.39
Powder content rate (%)	77.69	99.50	97.02	99.92

**Table 2 materials-13-04378-t002:** Concretes mix proportions.

Proportions	C20	C20-RCA30	C20-RCA50	C20-WPA30	C20-WPA50	C40	C40-RCA30	C40-RCA50	C40-WPA30	C40-WPA50
Cement	1.00	1.00	1.00	1.00	1.00	1.00	1.00	1.00	1.00	1.00
FNA	2.42	2.42	2.42	2.42	2.42	2.10	2.10	2.10	2.10	2.10
CNA	2.58	1.81	1.29	1.81	1.29	2.90	2.03	1.45	2.03	1.45
RCA	-	0.76	1.26	-	-	-	0.85	1.42	-	-
WPA	-	-	-	0.67	1.11	-	-	-	0.75	1.25
Water/cement ratio	0.60	0.50	0.50	0.50	0.50	0.40	0.40	0.40	0.40	0.40
Plasticizer (%)	0.760	0.368	1.000	0.563	0.750	0.750	0.750	0.750	0.750	0.750
Cement consumption (kg/m^3^)	354.91	367.97	367.97	367.97	367.97	388.88	388.88	388.88	388.88	388.88
Mortar rate (%)	57	57	57	58	59	52	52	52	53	53

**Table 3 materials-13-04378-t003:** Characteristic compressive strength of concretes obtained according to standards.

Concretes	f_cm_ ^a^	f_ck_ ^b^ NBR 6118:2014	f_ck_ ^b^ fib Model Code 2010	f_ck_ ^b^ Eurocode 2:2004	f_ck_ ^b^ ACI 318:2014
	(MPa)	(MPa)	(MPa)	(MPa)	(MPa)
C20	36.05	29.45	28.05	28.05	27.78
C20-RCA20	31.99	25.39	23.99	23.99	23.72
C20-RCA50	23.01	16.41	15.01	15.01	14.74
C20-WPA30	35.50	28.90	27.50	27.50	27.23
C20-WPA50	35.57	28.97	27.57	27.57	27.30
C40	49.71	43.11	41.71	41.71	40.80
C40-RCA30	47.48	40.88	39.48	39.48	38.78
C40-RCA50	38.65	32.05	30.65	30.65	30.75
C40-WPA30	41.39	34.79	33.39	33.39	33.24
C40-WPA50	56.25	49.65	48.25	48.25	46.75

^a^ The average strength obtained by results of six specimens; ^b^ the characteristic strength obtained by standards.

**Table 4 materials-13-04378-t004:** Elasticity modulus prediction equations proposed by standards.

Standard	Tangent Modulus (E_ci_)	Secant Modulus (E_cs_)
NBR 6118:2014	$E_{ci}=\alpha_{E}5600\sqrt{f_{ck}}$	$E_{cs}=\left(0.8+0.2\left(\frac{f_{ck}}{80}\right)\right)E_{ci}$
fib Model Code 2010	$E_{ci}=\alpha_{E}21,000\sqrt[{3}]{\frac{\left(f_{ck}+8\right)}{10}}$	$E_{c}=\left(0.8+0.2\left(\frac{f_{ck}}{88}\right)\right)E_{ci}$
Eurocode 2:2004	$E_{c}=1.05E_{cm}$	$E_{cm}=22{\left(\frac{f_{cm}}{10}\right)}^{0.3}$
ACI 318:2014	$E_{ci}=4700\sqrt{f_{c}^{\prime }}$	-

**Table 5 materials-13-04378-t005:** Tensile strength prediction equations proposed by standards.

Standard	Direct Tensile Strength (f_ct_)	Splitting or Indirect Tensile Strength (f_ct,sp_)
NBR 6118:2014	fctm=0.3(fck)23	$f_{ct,sp}=f_{ct}/0.90$
fib Model Code 2010	fctm=0.3(fck)23	$f_{ctm,sp}=f_{ct}/1.00$
Eurocode 2:2004	fctm=0.3(fck)23	$f_{ct,sp}=f_{ct}/0.90$
ACI 318:2014	-	fct=0.56(fck)12

**Table 6 materials-13-04378-t006:** Tested results of concretes mechanical properties.

Concretes	f_cm,cyl_	f_cm,cube_	f_ctm,sp_	E_cm_
	(MPa) ^a^ (COV) ^b^	(MPa) ^a^ (COV) ^b^	(MPa) ^a^ (COV) ^b^	(GPa) ^a^ (COV) ^b^
C20	36.05 (2.24%)	42.09 (5.48%)	3.99 (7.99%)	36.16 (5.25%)
C20-RCA20	31.99 (4.51%)	36.53 (5.14%)	3.10 (5.02%)	40.82 (3.59%)
C20-RCA50	23.01 (3.90%)	31.68 (6.86%)	2.64 (5.64%)	32.01 (4.79%)
C20-WPA30	35.50 (0.61%)	54.32 (4.62%)	3.57 (3.82%)	41.59 (3.97%)
C20-WPA50	35.57 (5.73%)	39.70 (5.63%)	3.31 (1.97%)	43.07 (5.80%)
C40	49.71 (3.40%)	54.16 (4.85%)	5.14 (4.98%)	38.11 (2.15%)
C40-RCA30	47.48 (5.72%)	48.41 (1.23%)	4.29 (5.86%)	43.89 (5.50%)
C40-RCA50	38.65 (15.47%)	41.91 (1.02%)	4.63 (2.19%)	36.03 (4.53%)
C40-WPA30	41.39 (10.44%)	43.78 (2.01%)	4.34 (4.72%)	46.85 (5.14%)
C40-WPA50	56.25 (7.30%)	58.78 (6.04%)	4.50 (0.57%)	44.03 (0.86%)

^a^ The concrete strengths in MPa; ^b^ the coefficient of variation in percent.

**Table 7 materials-13-04378-t007:** ANOVA analysis on compressive strength of all concretes.

Concretes		Average 1	Average 2	F	*p*-Value	F	Average Difference
		(MPa)	(MPa)				
C20	C20-RCA30	36.05	31.99	12.68	0.0200	7.71	4.06
C20	C20-RCA50	36.05	23.01	293.75	0.0001	7.71	13.03
C20	C20-WPA30	36.05	35.50	1.43	0.3200	10.13	0.55
C20	C20-WPA50	36.05	35.57	0.09	0.7800	7.71	0.47
C40	C40-RCA30	49.71	47.48	1.10	0.3400	6.61	2.23
C40	C40-RCA50	49.71	38.65	5.92	0.0900	10.13	11.06
C40	C40-WPA30	49.71	41.39	6.18	0.0900	10.13	7.92
C40	C40-WPA50	49.71	56.26	4.35	0.1700	18.51	6.54

**Table 8 materials-13-04378-t008:** ANOVA analysis on compressive strength of concretes.

Concretes	f_cm,cyl_	f_cm,cube_	F	*p*-Value	F	Average Difference
	(MPa)	(MPa)				
C20	36.05	42.09	11.74	0.0266	7.71	6.05
C20-RCA30	31.99	36.53	13.26	0.0149	6.61	4.54
C20-RCA50	23.02	31.68	55.36	0.0001	5.32	8.67
C20-WPA30	35.50	54.32	159.84	0.0001	6.61	18.82
C20-WPA50	35.57	39.71	6.53	0.0509	6.61	4.13
C40	49.71	54.16	4.51	0.1000	7.71	4.45
C40-RCA30	47.48	48.41	0.32	0.5900	5.99	0.93
C40-RCA50	38.65	41.91	0.53	0.5200	10.13	3.26
C40-WPA30	41.39	43.78	1.23	0.3200	6.61	2.39
C40-WPA50	56.26	58.78	0.55	0.5100	10.13	2.53

**Table 9 materials-13-04378-t009:** Compressive cylinder strength and compressive cube strength ratio of concretes.

Concretes	C20	C20-ARC30	C20-ARC50	C20-ARP30	C20-ARP50	C40	C40-ARC30	C40-ARC50	C40-ARP30	C40-ARP50
f_cm,cyl_ (MPa)	36.05	31.99	23.01	35.5	35.57	49.71	47.48	38.65	41.39	56.25
f_cm,cube_ (MPa)	42.09	36.53	31.68	54.32	39.70	54.16	48.41	41.91	43.78	58.78
f_cm,cyl_/f_cm,cube_	0.86	0.88	0.73	0.65	0.90	0.92	0.98	0.92	0.95	0.96

**Table 10 materials-13-04378-t010:** Measured splitting tensile strength and compressive strength ratio of concretes.

Concretes	C20	C20-ARC30	C20-ARC50	C20-ARP30	C20-ARP50	C40	C40-ARC30	C40-ARC50	C40-ARP30	C40-ARP50
f_cm,cyl_ (MPa)	36.05	31.99	23.01	35.5	35.57	49.71	47.48	38.65	41.39	56.25
f_ctm,sp_ (MPa)	3.99	3.10	2.64	3.57	3.31	5.14	4.29	4.63	4.34	4.5
f_ctm,sp_/f_cm,cyl_	11%	10%	11%	10%	9%	10%	9%	12%	10%	8%

**Table 11 materials-13-04378-t011:** ANOVA analysis on elasticity modulus of concretes.

Concretes		Average 1	Average 2	F	*p*-Value	F	Average Difference
		(MPa)	(MPa)				
C20	C20-RCA30	36.16	40.82	20	0.002	5.12	4.66
C20	C20-RCA50	36.16	32.01	15.38	0.004	5.12	4.15
C20	C20-WPA30	36.16	41.59	21.54	0.002	5.32	5.42
C20	C20-WPA50	36.16	43.07	27.19	0.001	5.12	6.9
C40	C40-RCA30	38.11	43.89	9.79	0.052	10.13	5.78
C40	C40-RCA50	38.11	36.03	2.6	0.248	18.51	2.09
C40	C40-WPA30	38.11	46.85	23.18	0.003	5.99	8.74
C40	C40-WPA50	38.11	44.03	85.82	0.012	18.51	5.92

**Table 12 materials-13-04378-t012:** Predicted tensile strength of concretes according to standards in comparison to measured results.

Concretes	Measured Results	Predicted Values							
	NBR 7222:2011	NBR 6118:2014		fib Model Code 2010		Eurocode 2:2004		ACI 318:2014	
	f_ctm,sp_ (MPa)	f_ct_ (MPa)	f_ct,sp_ (MPa)	f_ct_ (MPa)	f_ct,sp_ (MPa)	f_ct_ (MPa)	f_ct,sp_ (MPa)	f_ct_ (MPa)	f_ct,sp_ (MPa)
C20	3.99	2.86	3.18	2.77	3.08	2.77	2.77	ND	3.36
C20-ARC30	3.10	2.59	2.88	2.50	2.77	2.50	2.50	ND	3.17
C20-ARC50	2.64	1.94	2.15	1.83	2.03	1.83	1.83	ND	2.69
C20-ARP30	3.57	2.83	3.14	2.73	3.04	2.73	2.73	ND	3.34
C20-ARP50	3.31	2.83	3.14	2.74	3.04	2.74	2.74	ND	3.34
C40	5.14	3.69	4.10	3.61	4.01	3.61	3.61	ND	3.95
C40-ARC30	4.29	3.56	3.96	3.48	3.86	3.48	3.48	ND	3.86
C40-ARC50	4.63	3.03	3.36	2.94	3.26	2.94	2.94	ND	3.48
C40-ARP30	4.34	3.20	3.55	3.11	3.46	3.11	3.11	ND	3.60
C40-ARP50	4.50	4.05	4.50	3.98	4.42	3.98	3.98	ND	4.20

**Table 13 materials-13-04378-t013:** Predicted splitting tensile and compressive strength ratio of concretes.

Concretes	Measured Results	Predicted Values			
	f_ctm,sp_/f_cm,cyl_	NBR 6118:2014	fib Model Code 2010	Eurocode 2:2004	ACI 318:2014
		f_ct,sp_/f_ck_	f_ct,sp_/f_ck_	f_ct,sp_/f_ck_	f_ct,sp_/f_ck_
C20	11%	11%	11%	10%	12%
C20-ARC30	10%	11%	12%	10%	13%
C20-ARC50	11%	13%	14%	12%	18%
C20-ARP30	10%	11%	11%	10%	12%
C20-ARP50	9%	11%	11%	10%	12%
C40	10%	10%	10%	9%	10%
C40-ARC30	9%	10%	10%	9%	10%
C40-ARC50	12%	10%	11%	10%	11%
C40-ARP30	10%	10%	10%	9%	11%
C40-ARP50	8%	9%	9%	8%	9%

**Table 14 materials-13-04378-t014:** Predicted elasticity modulus of concretes according to standards in comparison to measured results.

Concretes	Measured Results	Predicted Values			
	NBR 8522:2017 E_cm_ (GPa)	NBR 6118:2014 E_ci_ (GPa)	Fib Model Code 2010 E_ci_ (GPa)	Eurocode 2:2004 E_ci_ (GPa)	ACI 318:2014 E_ci_ (GPa)
C20	36.16	36.46	39.56	33.94	24.77
C20-ARC30	40.82	33.86	38.02	32.74	22.89
C20-ARC50	32.01	27.22	34.06	29.66	18.04
C20-ARP30	41.59	36.13	39.36	33.78	24.52
C20-ARP50	43.07	36.17	39.38	33.8	24.56
C40	38.11	44.12	44.03	37.37	30.02
C40-ARC30	43.89	42.97	43.36	36.86	29.27
C40-ARC50	36.03	38.04	40.49	34.65	26.06
C40-ARP30	46.85	39.64	41.42	35.37	27.1
C40-ARP50	44.03	47.35	45.88	38.78	32.14

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
