# Peer review of "Modulus and Strength of Concretes with Alternative Materials"

_materials, 2020, doi:10.3390/ma13194378_

Round 1
Reviewer 1 Report
This article concerns a study with concretes produced with natural aggregates, recycled concrete aggregates (RCA) and porcelain waste (WPA). The topic is interesting and not much studied. The article exposes an adequate methodology, it is generally understandable, contains some errors to be fixed. Few aspects of the procedure used need to be further clarified to improve readability and overall quality.
In my opinion the document corresponds to the objectives of the magazine, if the newspaper is considered for publication by the publisher, it is necessary to make a minor revision:
- Abstract: Can be improved.
- Keywords: I suggest: concrete, natural aggregates, recycled concrete aggregates, waste porcelain aggregates, elasticity modulus
- Introduction: The state of the art is poor, although this subject has not been very studied. It is suggested to cite further papers as:
PEPE, M., TOLEDO FILHO, R.D.; KOENDERS, E.A.B.; and MARTINELLI, E.; (2014). Alternative processing procedures for recycled aggregates in structural concrete. Construction and Building Materials, 69, 124-132.
and papers dealing experimental test and reinforced concrete members as:
Guadagnuolo M., Donadio A., Tafuro A., Faella G., " Experimental Behavior of Concrete Columns Confined by Transverse Reinforcement with Different Details", The Open Construction & Building Technology Journal, 2020, 14: 250-265. DOI: 10.2174/1874836802014010250.
Watanabe, J. Niwa, H. Yokota and M. Iwanami, "Experimental Study on Stress-Strain Curve of Concrete Considering Localized Failure in Compression", J. Adv. Concr. Technol., Vol. 2, No. 3, pp. 395-407, 2004.
- From line 118: Check the bibliographic references [..]
- 3. Mechanical testing: I suggest to indicate the specimens number for each test type
- Tables 4 and 5: I suggest not to number the equations as they are never referred to in the text. In any case, review the equations template.
- Table 6: I ask you to comment on the values of fcm,cyl with high COV.
- Tables 7, 8 and 11: I ask to clarify how F and f values are calculated in the ANOVA analysis.
- Figures 8, 9, 10, 11, 12, 13: The unit of measurement of the y axis is missing.
- Line 297, 306, 315, 340, 369, 374: Do not start the sentence with the bibliographic reference.
- Tables 9 and 10: I suggest swapping x and y for lower tables.
- References: The required formatting is not applied.
Other comments are highlighted in the pdf file.

Author Response
Response to Reviewer 1 Comments
Reviewer 1: Comments and Suggestions for Authors
Reviewer 1: This article concerns a study with concretes produced with natural aggregates, recycled concrete aggregates (RCA) and porcelain waste (WPA). The topic is interesting and not much studied. The article exposes an adequate methodology, it is generally understandable, contains some errors to be fixed. Few aspects of the procedure used need to be further clarified to improve readability and overall quality.
Reviewer 1: In my opinion the document corresponds to the objectives of the magazine, if the newspaper is considered for publication by the publisher, it is necessary to make a minor revision:
Reviewer 1: Abstract: Can be improved.
Authors: The abstract was revised and some few changes were made to improve the text, such as:
Authors: Phrase before correction: This research aims to analyze the influence of selected recycled aggregates in the mechanical properties of moderate strength concretes and evaluate the difference between measured and predicted values of elasticity modulus.
Authors: Phrase after correction: This paper is about a research that analyzed the influence of recycled aggregates in the mechanical properties of conventional concretes and evaluate the difference between measured and predicted values of elasticity modulus.
Authors: Phrase before correction: Measured elasticity modulus were lower than modulus predicted by NBR 6118:2014 and fib Model Code 2010, while measured results were greater than values predicted by Eurocode 2:2004 and ACI 318:2014, as expected and it has indicated safety by the last two standards equations.
Authors: Phrase after correction: Measured elasticity modulus were lower than modulus predicted by NBR 6118:2014 and fib Model Code 2010, while measured results were greater than values predicted by Eurocode 2:2004 and ACI 318:2014, as expected and it has indicated safety by the last two cited standards.
Reviewer 1: Keywords: I suggest: concrete, natural aggregates, recycled concrete aggregates, waste porcelain aggregates, elasticity modulus
Authors: Keywords: Your suggest were done in text as highlighted in red. (line 34)
Reviewer 1: Introduction: The state of the art is poor, although this subject has not been very studied. It is suggested to cite further papers as:
Reviewer 1: PEPE, M., TOLEDO FILHO, R.D.; KOENDERS, E.A.B.; and MARTINELLI, E.; (2014). Alternative processing procedures for recycled aggregates in structural concrete. Construction and Building Materials, 69, 124-132.
Reviewer 1: and papers dealing experimental test and reinforced concrete members as:
Reviewer 1: Guadagnuolo M., Donadio A., Tafuro A., Faella G., " Experimental Behavior of Concrete Columns Confined by Transverse Reinforcement with Different Details", The Open Construction & Building Technology Journal, 2020, 14: 250-265. DOI: 10.2174/1874836802014010250.
Reviewer 1: Watanabe, J. Niwa, H. Yokota and M. Iwanami, "Experimental Study on Stress-Strain Curve of Concrete Considering Localized Failure in Compression", J. Adv. Concr. Technol., Vol. 2, No. 3, pp. 395-407, 2004.
Authors: The following sentence was inserted in line 51, considering one of the references suggested by the reviewer (PEPE, M., TOLEDO FILHO, R.D.; KOENDERS, E.A.B.; and MARTINELLI, E.; (2014). Alternative processing procedures for recycled aggregates in structural concrete. Construction and Building Materials, 69, 124-132): There was a study about a “ecological concrete” for structural use made by Brazil and Italy groups of researchers in 2004 [10].
Authors:The following sentence was inserted in line 53, considering one of the references suggested by the reviewer (Guadagnuolo M., Donadio A., Tafuro A., Faella G., " Experimental Behavior of Concrete Columns Confined by Transverse Reinforcement with Different Details", The Open Construction & Building Technology Journal, 2020, 14: 250-265. DOI: 10.2174/1874836802014010250): To relation the experimental values with predicted values from standards is always necessary to validate the research and the standards indication as [14] did with their columns with confined concrete.
Authors: The following sentence was inserted in line 302, considering one of the references suggested by the reviewer (Watanabe, J. Niwa, H. Yokota and M. Iwanami, "Experimental Study on Stress-Strain Curve of Concrete Considering Localized Failure in Compression", J. Adv. Concr. Technol., Vol. 2, No. 3, pp. 395-407, 2004.): According to [34], the stress-strain curves of concrete strongly depends on the aspect ratio of the concrete specimen.
Authors: The indicated papers were added in the references as numbers: [10], [14] and [34] and the other references were renumbered and corrected in the text.
Reviewer 1: From line 118: Check the bibliographic references [..]
Authors: In line 118, which became 121, the reference number was corrected.
Reviewer 1: 3. Mechanical testing: I suggest to indicate the specimens number for each test type –
Authors: In item 3, the number of specimens per test and by age of the concrete was explained as follows: Concrete specimens were cast while a vibrating table was used to compact the concrete. For each concrete mixture, thirty 100mm x 200mm cylindrical specimens and six 100mm cubic specimens were prepared. In particular, cylindrical specimens were used to measure the compressive strength (fcm,cyl) (6 specimens for 7 age days and 6 specimens for 28 age days), the elasticity modulus (Ecm) (6 specimens for 7 age days and 6 specimens for 28 age days) and the splitting tensile strength (fctm,sp) (6 specimens fo 28 age days), while cubic specimens were used only to test compressive strength at 28 age days and compare with values of compressive strength tested with cylindrical specimens.
Reviewer 1: Tables 4 and 5: I suggest not to number the equations as they are never referred to in the text. In any case, review the equations template.
Authors: The tables 4 and 5 were corrected to:
Table 4. Elasticity modulus prediction equations proposed by standards.
|
Standard |
Tangent modulus (Eci) |
Secant modulus (Ecs) |
|
NBR 6118:2014 |
Eci = aE * 5600 Ö(fck) |
Ecs = (0.8+0.2*(fck/80))Eci |
|
fib Model Code 2010 |
Eci = aE *21000((fck+8)/10)^(1/3) |
Ec = (0.8+0.2*(fck/88))Eci |
|
Eurocode 2:2004 |
Ec = 1.05*Ecm |
Ecm = 22*(fcm/10)^(0.3) |
|
ACI 318:2014 |
Eci = 4700 Ö(f’c) |
- |
Made by authors
Table 5. Tensile strength prediction equations proposed by standards.
|
Standard |
Direct tensile strength (fct) |
Splitting or indirect tensile strength (fct,sp) |
|
NBR 6118:2014 |
fctm = 0.3*fck^(2/3) |
fct,sp = fct/0.90 |
|
fib Model Code 2010 |
fctm = 0.3*fck^(2/3) |
fctm,sp = fct/1.00 |
|
Eurocode 2:2004 |
fct,m = 0.3*fck^(2/3) |
fct,sp = fct/0.90 |
|
ACI 318:2014 |
- |
fct = 0.56*fcm^(1/2) |
Made by authors
Reviewer 1: Table 6: I ask you to comment on the values of fcm,cyl with high COV.
Authors: The following sentence was added to line 266 - The individual specimens results of C40-RCA50 and C40-WPA30 was very dispersive, and was considered only 3 results of 6 specimens for the average and the COV calculation.
Reviewer 1: Tables 7, 8 and 11: I ask to clarify how F and f values are calculated in the ANOVA analysis.
Authors: The follow phrase was inserted in the text on line 274: In order to determine if the compressive strength of a concrete with recycled aggregates differs from its reference concrete, the ANOVA analysis is applied. Each row of Table 7 compares a pair of data. In these analysis results of p – value < 0.05 indicates the compressive strengths have a significant difference. When p – value >= 0.05 the test indicates the values are similar. F and f columns are ANOVA auxiliary variables. [30]
The test of C20 and C20-RCA30 indicates that their compressive strengths are statistically different (p – value = 0.0200 < 0.05) The average compressive strength reduced 4.06MPa due to the addition of recycled aggregates. On the other hand, the test of C40 and C40-WPA50 indicates the compressive strength are not significantly different (p – value = 0.17 > 0.05) Even though the average difference is 6.54MPa, the distribution of data are such that is t is not possible to determine any influence from the addition of recycled aggregates.
Reviewer 1: Figures 8, 9, 10, 11, 12, 13: The unit of measurement of the y axis is missing.
Authors: The unit MPa was inserted in figures 8, 9 e 11 and the unit GPa was inserted in figures 10, 12 e 13.
Reviewer 1: Line 297, 306, 315, 340, 369, 374: Do not start the sentence with the bibliographic reference.
Authors: It was adjusted in lines 291 and 293 as follow: It was assumed by [31] that it is due to the chemical and physical similarities between waste porcelain aggregates and natural aggregates. It was also state by [32] that porcelains have high .....
Authors: In line 309 follows: It was appointed by [36] that the effects, however, ……
Authors: In line 327 follows: It was said by [37] that normally specimens….
Authors: In line 329 follows: It was explained by [39] that a possible motive would be the effect ……
Authors: In line 353 follows: It was also state by [35] that compressive and tensile strength are closely related ……
Authors: In line 383 follows: It was showed by [41] that as RCA replacement rate increases, ……
Authors: In line 388 follows: It was also observed by [42] an increase in the modulus of elasticity of self-consolidating concrete
All frases above is highlighted in yellow in the text.
Reviewer 1: Tables 9 and 10: I suggest swapping x and y for lower tables.
The suggestion was accepted and the tables were redone, highlighted in yellow in the text
Reviewer 1: References: The required formatting is not applied.
Authors: The references have been adjusted to the requested format. The DOI address has been added to references whose existing address is highlighted in yellow.
Correction indications in the paper by the file (Reviewer 1):
Authors: The numbers corresponding to the works of the cited authors were corrected, the numbering format of equation (1) was corrected, the unit of elasticity module from MPa to GPa was corrected in table 6 and the mean reference resistance value of 41.21 MPa to 49.71MPa in table 7 and the other values were checked.

Reviewer 2 Report
The paper investigates modulus and strength of concretes with alternative materials: concretes with natural aggregates, recycled concrete aggregates (RCA) and waste porcelain aggregates (WPA). The research concerns the influence of recycled concrete aggregates and waste porcelain aggregates into mechanical properties of 20MPa and 40MPa strength class concrete, to verify if there are any similarities between the predicted values calculated by different standards, to compare the predicted values with measured results and to evaluate the compressive strength of concretes cast in cylindrical and cubic specimens. To validate the results, an ANOVA analysis was performed. An ANOVA analysis was also performed to verify whether the change in geometry of the specimens affects the compressive strength of the studied concretes.
The subject of the paper is interesting, but some major revision should be considered.
Please see remarks below.
(1) Please reorganize and develop the references in the introduction section to make it more coherent and logical about this specific topic described in the paper.
(2) In the introduction the authors had mentioned some related papers in the literature. Could you please provide more comments to show the advantage and progress of the present researches in this study over previous works.
(3) What is the novelty of the paper?
(4) Page 19, line 508, Authors state: “Most researches about the usage of recycled aggregates are concerned with the influence of these aggregates on compressive strength.But to ensure good quality structural concrete, it is important to investigate other parameters such as the elasticity modulus and tensile strength”. This remark is obvious and adds nothing new.
Author Response
Response to Reviewer 2 Comments
Reviewer 2: (1) Please reorganize and develop the references in the introduction section to make it more coherent and logical about this specific topic described in the paper.
Authors: Phrases were inserted inside Introduction and other existent phrase were changed places, as highlighted in text.
Reviewer 2: (2) In the introduction the authors had mentioned some related papers in the literature. Could you please provide more comments to show the advantage and progress of the present researches in this study over previous works.
Authors: It was inserted the follow phrase: Thus, this work, in contrast to the works mentioned above, showed the possible use of national and international standards for structural projects, with conventional strength concrete, made with certain recycled aggregates.
Reviewer 2: (3) What is the novelty of the paper?
Authors: This paper shows that, for some recycled aggregates and for conventional strength concrete, is possible to use the same standards for structural design.
Reviewer 2: (4) Page 19, line 508, Authors state: “Most researches about the usage of recycled aggregates are concerned with the influence of these aggregates on compressive strength. But to ensure good quality structural concrete, it is important to investigate other parameters such as the elasticity modulus and tensile strength”. This remark is obvious and adds nothing new.
Authors: Ok.

Round 2
Reviewer 2 Report
The required corrections were made.